# Automated Defect Detection and Decision-Support in Gas Turbine Blade Inspection



**Jonas Aust** [1,*] **, Sam Shankland** [2]**, Dirk Pons** [1] **, Ramakrishnan Mukundan** [2] **and Antonija Mitrovic** [2]

[1] Department of Mechanical Engineering, University of Canterbury, Christchurch 8041, New Zealand; dirk.pons@canterbury.ac.nz

[2] Department of Computer Science and Software Engineering, University of Canterbury, Christchurch 8041, New Zealand; sjs227@uclive.ac.nz (S.S.); mukundan@canterbury.ac.nz (R.M.); tanja.mitrovic@canterbury.ac.nz (A.M.)

* Correspondence: jonas.aust@pg.canterbury.ac.nz; Tel.: +64-210-241-3591

**Abstract:** Background—In the field of aviation, maintenance and inspections of engines are vitally important in ensuring the safe functionality of fault-free aircrafts. There is value in exploring automated defect detection systems that can assist in this process. Existing effort has mostly been directed at artificial intelligence, specifically neural networks. However, that approach is critically dependent on large datasets, which can be problematic to obtain. For more specialised cases where data are sparse, the image processing techniques have potential, but this is poorly represented in the literature. Aim—This research sought to develop methods (a) to automatically detect defects on the edges of engine blades (nicks, dents and tears) and (b) to support the decision-making of the inspector when providing a recommended maintenance action based on the engine manual. Findings—For a small sample test size of 60 blades, the combined system was able to detect and locate the defects with an accuracy of 83%. It quantified morphological features of defect size and location. False positive and false negative rates were 46% and 17% respectively based on ground truth. Originality—The work shows that image-processing approaches have potential value as a method for detecting defects in small data sets. The work also identifies which viewing perspectives are more favourable for automated detection, namely, those that are perpendicular to the blade surface.

**Keywords:** automated defect detection; blade inspection; gas turbine engines; aircraft; visual inspection; image segmentation; image processing; applied computing; computer vision; object detection; maintenance automation; aerospace; MRO

## 1. Introduction

Aircraft engine maintenance plays a crucial role in ensuring the safe flight state and operation of an aircraft, and image processing—whether by human or automatic methods—is key to decision-making. Aircraft engines are exposed to extreme environmental factors such as mechanical loadings, high pressures and operating temperatures, and foreign objects. These contribute to the risk of damage to the engine blades [1–4]. It is vitally important to ensure high quality inspection and maintenance of engines to detect any damage at the earliest stage before it propagates towards more severe outcomes. Missing defects during inspection can cause severe damage to the engine and aircraft and have the potential to cause harm and even fatalities [5–7].

There are several levels of inspection, each with their own tools and techniques. A comprehensive inspection workflow is presented in Figure 1. The V-diagram shows the different levels and their hierarchy. The more detailed the inspection (further down in the diagram), the better the available inspection techniques, but at the same time the higher the cost introduced for further disassembly and reassembly. The different inspection levels can be summarised into two main types of inspection: in-situ borescope inspection performed on-wing or during induction inspection and subsequent module and piece-part

inspection where the parts are exposed. While borescope inspection is an essential first mean of inspection to determine the health and condition of the parts and subsequently make the decision as to whether further disassembly and detailed inspection is required, it also has limitations of relatively low image quality and poor lighting conditions inside the engine [8]. Furthermore, there is limited accessibility, which creates the need to use different borescope tips, which in turn leads to a high variation of images [9]. Due to the challenging borescope inspection environment, this research focuses on piece-part inspections, where these conditions can be better controlled. If successful, the proposed method could be refined and might then be transferable to higher levels of inspection, namely, module and borescope inspection.

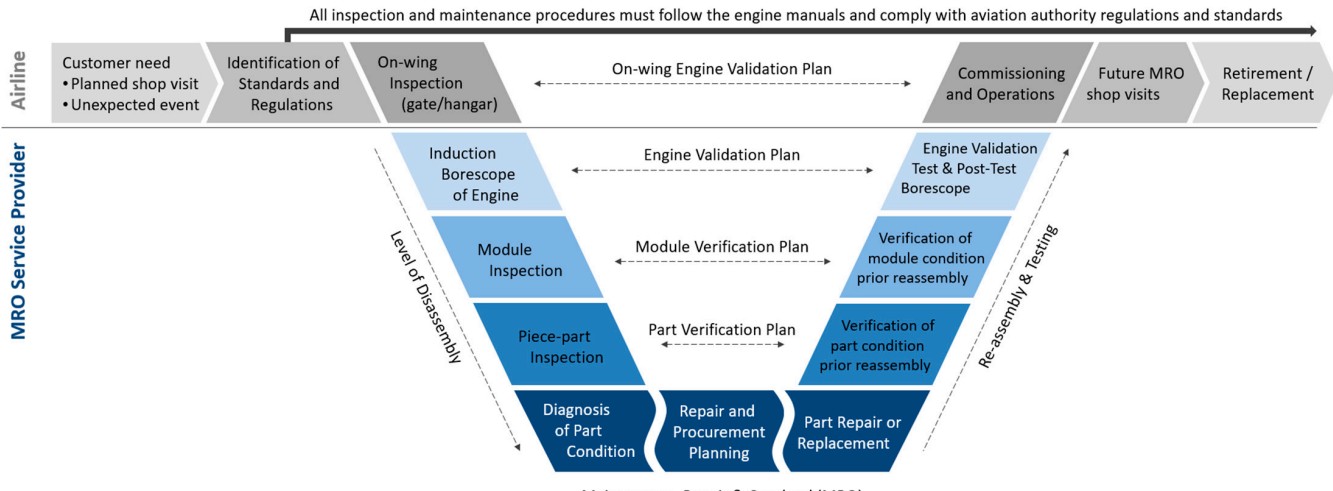

**Figure 1.** Inspection workflow and hierarchy of levels of inspection.

During all visual inspections, a skilled technician obtains an appropriate view of the part and evaluates the condition by searching for any damages. When a defect is detected, the inspector has to make a decision whether or not it is acceptable, i.e., check if it is within engine manual limits. This decision is based on the inspector's experience and to some extent on the risk appetite. Some inspectors tend to take a more risk adverse stand, which may lead to a costly tear down of the engine or scrapping of airworthy parts. Both tasks, defect detection and evaluation, are time consuming and tedious processes that are prone to human error caused by fatigue or complacency. This entails the risk of missing a critical defect during inspection. Thus, there is a need to overcome those risks and support the human operator, while improving the inspection quality and repeatability, and decreasing the inspection times. Ultimately, this has the potential to improve aviation safety through reduction of accidents in which defects were missed during the inspection task [10,11].

In this research, the focus is on defects present on the leading and trailing edges of compressor blades. These blades are located as per Figure 2 and highlighted in yellow. An isolated blade is presented next to it.

The most common edge defects are dents, nicks and tears. An overview of the defect types and their characteristics together with a sample photograph is shown in Figure 3 below. It should be noted that the sample images show severe defects. This is for demonstration purposes only, to highlight the difference between the different defect types. The test dataset also contained blade images with smaller defects that are more difficult to detect. Detecting those defect types is important as they can lead to fatigue cracks [13] resulting in material separation and breakage of the entire blade under centrifugal load [14], which has the potential to cause severe damage to the engine and aircraft.

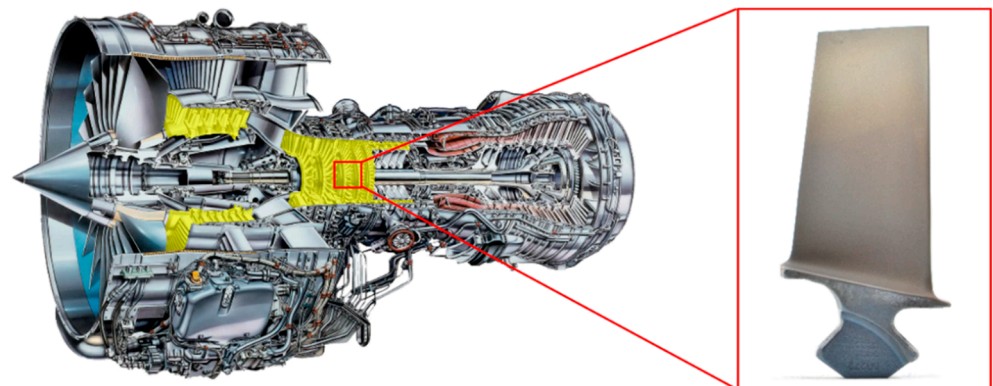

**Figure 2.** Cross section of a V2500 gas turbine engine with highlighted compressor blades in yellow and exposed sample. Image adapted from [12].

| Defect Type | Characteristics | Sample Image |
|---|---|---|
| **Dent** | A small, smooth indention with rounded edges, corners and bottom caused by mechanical impact of a dull object. Often, dents can be found at or close to the blade edges. | |
| **Nick** | A small, sharp cut on the edges of a blade caused by a striking object. A nick has a characteristic V-shaped bottom, and concentrates stresses. | |
| **Tear** | Separation of material by tensile stresses imposed by a sharp object. It is apparent by ragged or irregular edges. | |
| **Non-defective** | Blade that may show some deposits, but has otherwise no visually detectable damage. | |

**Figure 3.** Overview of edge defects and their characteristics (Taxonomy adapted from [15]).

## 2. Literature Review

### 2.1. Automated Visual Inspection Systems (AVIS)

In aviation, automated inspection systems have been developed for detecting damages on aircraft wings and fuselage [16–21], tyres [22], engines and composite parts [16,23–26]. Several airlines have shown increasing interest in automating visual inspection and tested several systems, including inspection drones for detection of lightning strikes, marking checks and paint quality assurance [27,28], robots with vacuum pads for thermographic crack detection [29] and visual inspection robots for fuselage structures [30]. Engine

parts such as shafts [31,32], fan blades [20,33], compressor blades [9,34,35] and turbine blades [9,26,36–43] are also of particular interest as all are safety-critical to flight operations.

Deep learning is used in several types of classification problems where feature representations are automatically learned using large amounts of labelled training data. These models can have many, possibly millions of, trainable parameters. In contrast, the problem at hand deals with simple geometrical characteristics of images that require few parameters (e.g., slope of a line segment). Such problems can be solved directly using a small set of image processing methods that extract and quantify relevant features. A small feature space also makes it possible to use simple rule-based detection and classification algorithms. Deep learning based solutions could still be attempted, where the entire blade image is provided as input and classified into one of the defect types.

One drawback of these techniques, however, is the need for large training datasets in order to produce good models. This is potentially problematic in the case of jet engines, as there are many different parts with variations in geometry. In particular, the compressor and turbine blades have different geometric features in addition to size changes. There are also the stationary vanes between each row of moving blades. All this variety adds up to a formidable detection task for AI systems, and hence is still the preserve of expert human inspectors. Humans have the ability to understand the context of what they are looking at, specifically what is and is not important in the visual field.

Several attempts to overcome the challenge with small datasets have been made, including the approach developed by Kim et al., which was able to detect nicks with a 100% accuracy on training data [43]. The approach used the scale invariant feature transform (SIFT) algorithm [44,45] and principal component analysis to produce a damage representation, which is then compared with input images. Should a sufficient level of feature matching be achieved, the image was processed by a CNN to provide a classification. Although it was able to achieve a high detection rate, the software was only able to detect nicks, no other defect types.

Other research using CNN techniques have explored detection of cracking in wind turbine blades [46], cracking on the surfaces of compressor blades in jet engines [9] and detection of surface erosion on turbine blades in jet engines [47]. All of these approaches use some form of feature extraction or segmentation techniques to normalise the input into a trained CNN and typically achieved a high accuracy of detection. However, the training examples were of advanced damage that is clearly visible to the human eye. In reality, a system needs to detect defects at much smaller scales of severity, and this has not yet been convincingly demonstrated in the literature. Additionally, all these applications required large quantities of training data. The issue, as identified by Wang using x-ray inspection and CNN, is that lack of training data for rare defects results in extremely poor performance of the network when exposed to novel defects [48].

These methods use feature extraction techniques and then classification of the features in comparison to features trained from both damaged and non-damaged blades. Of the image-processing algorithms, positive results have been shown for bilateral filtering and Gaussian blur algorithms [49,50].

However, the range of defects that can be detected is still limited. Typically, neural networks are used for classification tasks [51]. However, they require significant amounts of data to accurately perform a classification, especially with increasing number of classes [52–54]. There exist several commercial AI software for inspection of gas turbine blades. Some focus on borescope inspection [55,56], while others target the automated inspection of piece-parts [57–59]. They all use Deep Learning AI, which is perhaps feasible due to their fortunate commercial situation of being able to collect a large dataset of defective blade images. Thus, there is a fair chance that Deep Learning AI can be successfully applied, in the right conditions. However, in cases where images are scarce, the neural network approach may have inherent limitations due to the variety of defect types and the rarity of some defects. Consequently, there is value in exploring other approaches, especially those that are less critically dependent on large datasets.

## 2.2. Automated Defect Measurement

Before a decision can be made whether a defect is within or out of limits, the damage has to be measured. The biggest challenge with measuring the defect size is the variety of defect shapes and appearances. Different attempts have been made for estimating the defect size. For example, borescope instruments allow measuring the defect size using stereo imaging or optical phase shifting. However, this is a manual and time-consuming task, since the inspector has to acquire an image first and then mark the contour points (start and end of defect) between which the distance shall be measured. The situation can be improved by providing a scaled measurement grid [60].

In automatic visual inspection systems, traditional approaches use the bounding box or the horizontal cross-section of the detected defect to estimate the defect size [61,62]. For surface defects, this does not represent the actual defect size and thus they recommend using the largest dimension of all cross-sections of the detected defect [62]. Those authors developed a software tool to detect and to calculate the defect characteristics, including the defect size, shape, location and type. However, it is not mentioned which methods are used to extract the defect information from the image and how the defect size is estimated. The only information given is that the maximum length across all cross-sections was used. This however does not apply to engine blade inspection, where the engine manual limits determine the serviceability based on the depth of the defect independent of its other dimensions, such as defect height or volume of missing material. The defect depth is not always the largest dimension, and thus, the width of the bounding box provides a better estimate of the critical defect size than the maximum length of all cross-sections.

Most surface defects appear in a circular or elliptical shape rather than a rectangle. Hence, there has been work to approximate the real defect shape [63,64]. Volume-based measuring methods have been attempted [65], though is less suitable for edge defects, where material is deformed or missing, and thus, there is no depth to measure.

## 2.3. Decision-Support Systems for Maintenance and Inspection Applications

In the literature, there are mainly three types of decision support systems: (1) maintenance and inspection decision support systems for selecting the best inspection techniques and timing for performing a maintenance cycle [66–70], (2) decision support after the inspection is performed to determine if the findings are critical [71] and (3) a combination of both with recommendations for the best repair action based on the findings [72]. The focus of this paper is on the decision support after the inspection is performed and the relevant literature is reviewed in this section.

The work by Zou et al. [72] proposed a support tool to improve the inspection, maintenance and repair decision-making, taking into account factors that affect the defect propagation. The approach was based on risk assessment and life cycle cost analysis. The decision support tool provided answers to the questions where to inspect, how frequent to inspect, and what technique to use. Furthermore, it recommended whether, when and how to repair the defects.

While they used the risk of failure, this is less relevant in aircraft engine maintenance, Instead the risk is incorporated in the engine maintenance limits of allowed defect sizes and thus does not need to be included in the decision support tool.

In the medical field, a skin inspection system was developed to search for pigments that indicate skin cancer [71]. The software utilised image-processing techniques, such as threshold-based feature segmentation. After the detection of skin anomalies, a machine learning based decision support tool was introduced to help the classification of those findings and determine whether the anomaly was benign lesions or melanoma. It took into account several influence factors that have an effect on the likelihood of melanoma such as gender, age, skin type, and affected area of the body. The concept that different body parts have different risk levels can be translated to engine blade inspection, where the blade has different tolerance areas as well. This tool used machine learning. Due to the scarce amount of data, machine learning might be less suitable for a decision support tool

in the MRO domain. Furthermore, the approach by Alcon has only two classes and the threshold is determined based on the training dataset. In the case of blade inspection, the threshold is determined by well-defined engine manual limits.

*2.4. Gaps in the Body of Knowledge*

Although the field has moved towards automated visual inspection in the maintenance environment, there are limitations. Typically, neural networks are used for defect detection and classification [73].

It is generally accepted that the performance of a deep learning neural network improves logarithmically with increasing sample size [74–77]. Several sources state that a dataset of 1000 images per class is required to successfully train a neural network [74,78–80]. A recent example in the medical field for COVID-19 detection used a dataset of 1128 images [81]. Sun et al. [77] used 300 million images to train a neural network. The number of images may be reduced by using a pre-trained model (where applicable) and smart augmentation to artificially create a larger dataset. Good accuracies with reasonable classification results can be achieved with sample sizes as small as 150–500 images per class [76]. However, "good" and "reasonable" are subjective. It can be summarised that a "small" dataset for neural networks is at least 150 images, though about a thousand images is the norm, and "big data" comprises millions of records. In contrast, the minimum number of images for traditional image processing approaches is much less, of the order of about 10–100 [82,83].

Hence, the neural network method critically depends on relatively larger training and test datasets compared to image processing methods [52–54]. Furthermore, there are several sizes, shapes and types of blades (especially compressor versus turbine differentiation), which further increases the required amount of data. This limitation is also prevalent in more advanced neural networks, such as CNNs and their variants. Thus, there is a need to develop defect detection system that would perform well for small datasets and rare defects. The rare defects are precisely the types that are important to detect.

An alternative to neural networks and their variants comes in the form of classical image processing techniques that have been used in the field of computer vision for a long time [84]. There is a lack of recent applications of these techniques to blade inspection. In fact, the field is somewhat weak and has been dominated by the neural networks and deep learning approaches instead [9,19,47,85].

Furthermore, most research focuses on defect detection on turbine blades rather than compressor blades. This encompasses mainly crack detection [9,33,37,38,42], as this is the most critical type of defect and the main source of failure. Nonetheless, other types of defects can lead to significant shortage of the part life cycle and propagate towards cracks leading to the same consequences. For example, in the compressor stage, the blades are vulnerable to impact damage on their leading edges, in the form of small nicks and dents. Broken blades propagate through the engine, damaging downstream parts. Thus, detecting small damages at the front of the engine is particularly important. Hence, there is a need to find those defects that are poorly represented in the literature, such as nicks and dents in the compressor blades.

A pervasive problem is that many systems presented in the literature have been developed on samples with obvious defects that would quickly be detectable by any trained human operator, and hence do not need support. The smaller defects are harder to detect and hence a smart inspection system could have benefit. Furthermore, the reviewed systems have difficulty detecting multiple defect types. Most have focused on detecting cracks on turbine blades, since these are highly critical. However, other types of defect are of similar importance or even more critical, e.g., tears or broken-off material. In practice, it is of utmost importance to detect all defects that are critical and have the potential of negatively affecting flight operation.

The accurate detection of blade condition early in the maintenance cycle is essential. False positives can commit the engine to an unnecessary expensive remanufacturing pro-

cess. False negatives on the other hand may cause a continued operation of the engine in a defective state with the potential loss of the entire aircraft and passengers. Consequently, decisions made while the engine is still on wing have a material impact on the organisational risks and human safety. Detecting a defect is only the first step, while the subsequent decision whether the defect is acceptable has a bigger impact on the operation. Most literature on decision support systems in the maintenance domain focused on preventive and predictive models to forecast and prioritise maintenance activities [70]. However, little to no attention has been directed to the maintenance actions after the inspection has been performed. No maintenance decision support tool appears to exist, neither in the aviation industry nor in the journal literature, which takes into account engine manual limits as a basis for the decision. The present paper specifically addresses this problem.

## 3. Methods

### 3.1. Purpose

The purpose of this research was to develop software with two main functions. The first one is the automated detection of blade defects in the aero engine domain. This comprises the detection and location of the defect, and quantitative assessment of the defect morphology, including the defect size measured in height, depth, and area of missing material, and the edge deformation measured in change of angle. The scope is limited to the detection of edge defects, rather than airfoil defects. This was because the edge defects are more important from a safety perspective, since this is where cracks and other catastrophic failures originate. In contrast, surface defects lead to efficient losses, but no further damage or harm. An image perspective comparison was made to determine the best view with the highest detection accuracy.

The second function is a decision-support tool to assist the inspector by providing a recommended maintenance action based on a comparison of the defect findings (from the previous detection software) and the limits extracted from the maintenance manual. The potential benefit of this is shortened inspection times, while improving the detection accuracy and thus quality.

### 3.2. Approach

The overall approach comprised (1) image acquisition, (2) development of a detection software and (3) development of a decision support tool. Both (2) and (3) use heuristics. The overall structure of the solution is shown in Figure 4 and will be further discussed in the following sections.

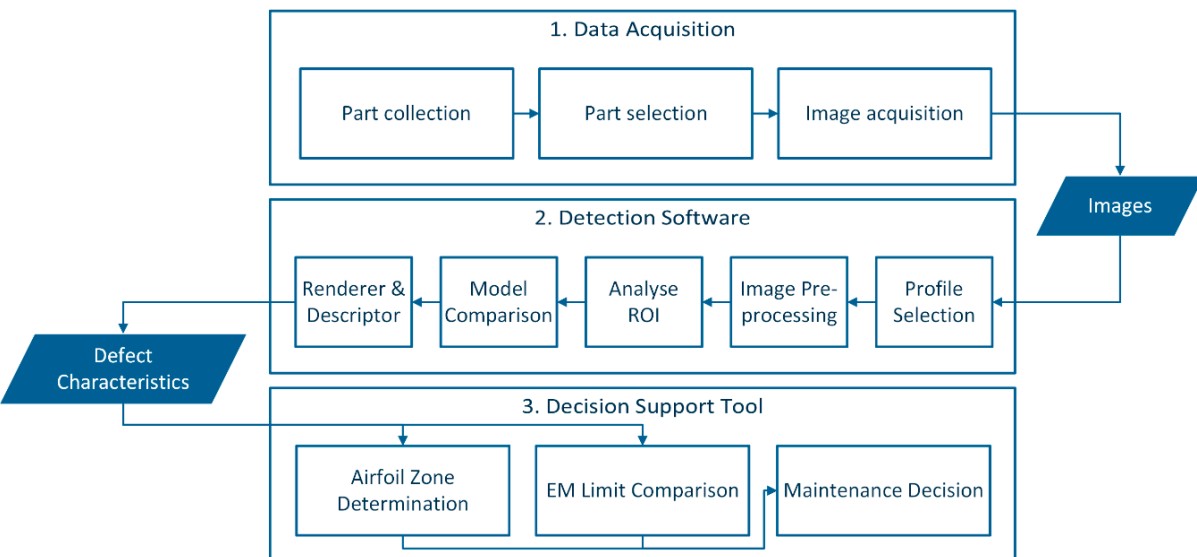

**Figure 4.** Overview of research approach.

### 3.2.1. Data Acquisition

The research sample for model generation and testing contained a mix of 52 damaged and 28 non-damaged high-pressure compressor (HPC) blades of stages 6 to 12 from V2500 aircraft engines undergoing maintenance. This dataset is small in comparison to the related work introduced in the literature review. The blades were in different dirty conditions. There are two main categories of defects, namely edge defects and surface defects. Edge defects typically appear a change in shape of the leading or trailing edge. Surface defects in turn, appear as a change in gradient. This work focuses on edge defects, as these are more critical due to their inherent risk of propagating and cause severe engine damage if they stay undetected. The most common defect types in this category are nicks, dents, and tears on leading and trailing edges. The defect proportion of the research sample was 42% nicks, 38% dents and 20% tears. Only blades with defects that are visually detectable were used as the detection software as well as the human eye of the operator performs an optical analysis.

A standardised image acquisition procedure was developed to ensure repeatability. This includes eight standardised camera views and a defined camera setup. The setup comprises a self-built light tent with three ring-lights (Superlux LSY 6W LED) and a 24.1 mega pixel Nikon D5200 DSLR camera with Nikon Macro lenses (AF-S Micro Nikkor 105 mm 1:2.8 G) mounted on a tripod (SLIK U9000). The acquired images were stored in JPEG format with a resolution of 4928 × 3264 pixels. This setup was chosen, as we wanted to represent an ideal environment for on-bench piece-part inspection. In total, 80 blade samples were collected and images thereof acquired, before they were submitted to the detection software.

Typical levels of defects in blades are shown in Figure 5. The sample blades originated from an engine with foreign object damage (FOD), and thus, the defects represent an intermediate level of damage.

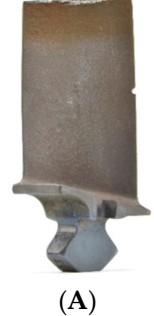 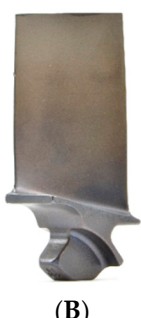 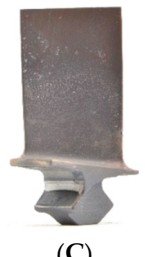 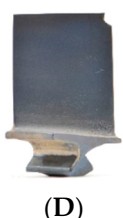 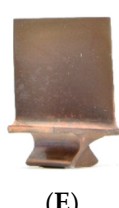

| (A) | (B) | (C) | (D) | (E) |

**Figure 5.** Sample blade defects: (**A**) nick on leading edge, (**B**) dent on trailing edge, (**C**) nick on leading edge, (**D**) teared-off corner, (**E**) dent on leading edge.

For 33 blades, images from eight different perspectives were taken, giving 264 images. The remaining 47 blades were photographed from perspectives P3 and P7 (perpendicular to the airfoil) providing a dataset of 94 images. As shown in Figure 6, the different blade perspectives represent the rotation of the blade in 45-degree increments.

### 3.2.2. Detection Software

As identified above, the approach taken here eschewed neural networks and rather focussed on image processing. The detection software was developed in Python version 3.7.6 [86] using the OpenCV library version 4.3.0 [87]. It involves a series of algorithms applied to generate a ground-truth model of an undamaged blade and subsequently processed each input image to detect edges. The principle of detection was based on breaks in line continuity and acts as the implemented heuristic function. The algorithm parameters were determined using an iterative approach to optimise the performance of the model as described in more detail in the following sections.

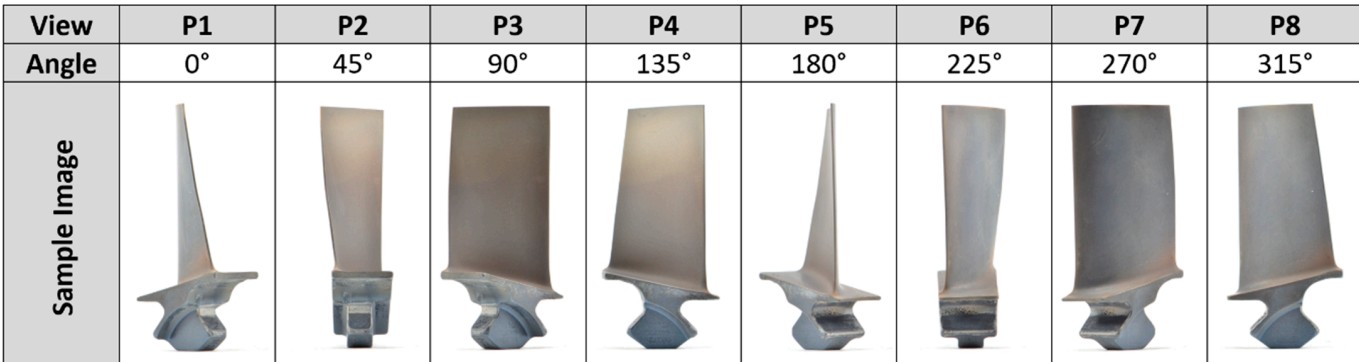

| View | P1 | P2 | P3 | P4 | P5 | P6 | P7 | P8 |
|------|----|----|----|----|----|----|----|----|
| Angle | 0° | 45° | 90° | 135° | 180° | 225° | 270° | 315° |

**Figure 6.** Different blade perspectives.

First, 20 non-defective blades were used to establish the ground truth model with respect to the heuristic function. Next, JPEG images were imported and processed following a specific procedure. This included image pre-processing to reduce the noise, converting the image to greyscale and compressing the image size down to 30% to improve the performance (computation speed).

Thereafter, regions of interest (ROI) were generated on the input image using the same heuristic function. The ROIs were then compared region by region to the ground-truth model and any significant difference between the two was considered as defective area. This area was marked on top of the input image by the renderer in form of a bounding box around the detected area. Finally, the descriptor performed an analysis of the detected regions and calculated their mathematical properties as described in the following sections. These defect characteristics were then exported together with the marked image as an output file.

The system architecture is shown in Figure 7. Please note that the surface defect heuristic highlighted in red was not implemented; however, it acts as an example of adding additional heuristics for characterisation of different defect types.

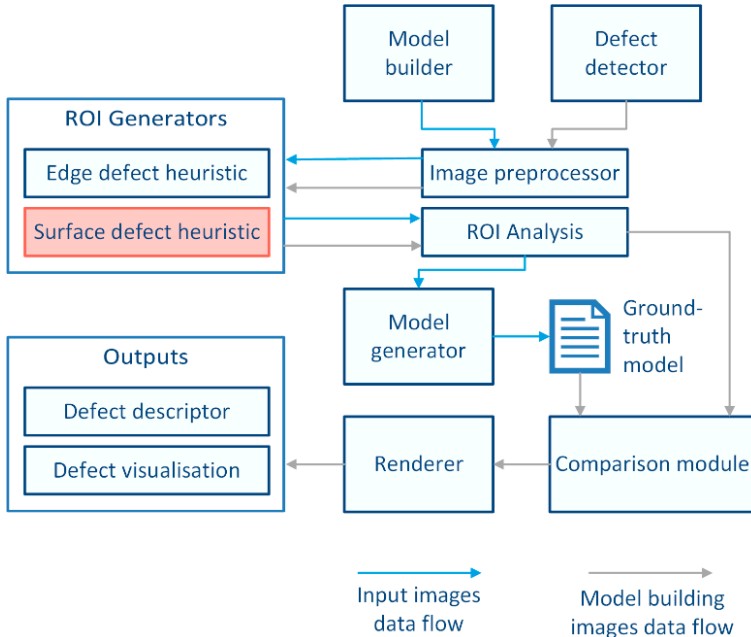

**Figure 7.** Architecture diagram of the detection system.

Image Processing

The image processing operation is as follows. The input image is stored in a matrix of the size $(H, W, C)$, whereby $H$ represents the height and $W$ the width of the input image.

*C* stores the colour channels Red, Green and Blue of the image. This matrix is then scaled down on the *H* and *W* axes, and the *C* axis is collapsed to 1 as the image is converted to grayscale. The grey-scaling follows the equation as specified by OpenCV:

$$Y = 0.299 \times R + 0.587 \times G + 0.114 \times B \tag{1}$$

with *R*, *G*, *B* representing the colour channels red, green, and blue respectively. The matrix of size $(H, W, 1)$ is then convolved with a bilateral filter kernel (shift-invariant Gaussian filter) to produce a de-noised image.

Generation and Analysis of Regions of Interest

All processed blade images are passed to the ROI generator, which applies the heuristic function that generates points of interest. In this case, the heuristic is based around finding breaks in line continuity. To do so, the edges of the blade are found using the Canny edge detector. As all blade images contain one foreground object against a bright and uniform background, effective background segmentation and edge detection in such images can be achieved by using adaptive thresholding methods that provide robustness against illumination variations. Commonly used lower and upper threshold values are certain percentages (empirically determined) of mean, median or Otsu thresholds [86]. In the proposed method, the lower and upper thresholds used for the Canny algorithm are 0.66 and 1.33 M, respectively, where M is the median pixel intensity.

The Suzuki algorithm [88] is then applied to the found edges in order to extract contours and order them in a hierarchical structure. The external contours are placed at the top of the hierarchy; in this case, these are the contours relating to the outside of the blade. Internal contours are discarded, as they are not relevant, since they represent the contours of surfaces on the blade. The points in the external contours are concatenated forming an array of points representing the contour of the entire outside of the blade. This point array is iterated through to find the differences in angles $\Delta\theta$ between two consecutive line segments along an edge contour using the inverse tangent extension function $atan2$ as shown in Equation (2):

$$\Delta\theta = \left| atan2\left(c_y - b_y,\ c_x - b_x\right) - atan2\left(b_y - a_y, b_x - a_x\right) \right| \tag{2}$$

where *a*, *b* and *c* are the points in which the angle difference is computed (Figure 8). These values are reassigned to new points in the point array as it is iterated through. Should the threshold of $\frac{\pi}{12} rad$ be exceeded for $\Delta\theta$, the points *a*, *b* and *c* are added to a suspect points set. The threshold value was selected using an imperative approach such that the impact of noise at the edge of the blade was minimised whilst retaining high accuracy in detecting derivations from the continuity of the edge contour. We experimented with various values ranging from zero to $\frac{\pi}{6} rad$ in $\frac{\pi}{180}$ increments using a sample blade and determined that the best result was achieved with $\frac{\pi}{12} rad$. Contour following was used because the defect types that are being detected exhibit the common characteristic of having non-contiguous or sharp changes in the direction of the contour on the edge of the blade.

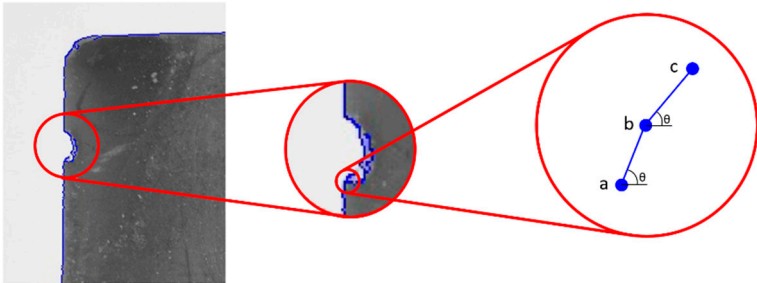

**Figure 8.** Shows the points in the contour (blue) and the determination of the angle between points.

In order to create regions of interest, the bounding box of the edge contour is found and subdivided into $R \times R$ regions. The proposed method used $R = 10$ in order to produce 100 regions on the blade. Each point in the suspect point set is then assigned to a region based on the $x, y$ coordinate of the point being located within the bounds of that region. Each region is assigned an index $R_{x,y}$, which determines the location in relation to the top-left corner of the bounding box.

Model Generator

The model generator module takes a set of non-damaged blades and performs the edge defect heuristic method in order to determine the ground truth. This produces a 4D array of size $(N, X, Y, P)$, where $N$ is the number of example images, $X$ and $Y$ are the indices of the region, and $P$ is the suspect point list. Then the average density of each region is calculated using the number of points per region to produce a matrix of size $(R, R)$, where $R$ is the number of axis divisions for each region. This matrix is then stored as the ground truth model for the edge defect heuristic.

Comparison Module

The comparison module is used to load a built model and compare the ROI analysis of the input image and the ground truth model. The comparison is threshold based, in which a region in the input image that has a density that is greater than a certain multiplier of the model for that region will be marked as defective, per Equation (3):

$$Defect_{x,y} = \begin{cases} 1 & Input_{x,y} > Model_{xy} \times multiplier \\ 0 & otherwise \end{cases} \tag{3}$$

$Defect_{x,y}$ is a True/False value of the defectiveness for the region with index $(x, y)$. $Model_{x,y}$ is the value in the model for the same region index $(x, y)$ and $Input_{x,y}$ is the number of suspect points in the input image. The multiplier was determined by using a 1D-grid search and selecting a value from the range of 1 to 15 that produced the best F1 score. The lower bound of 1 was selected as it was expected that a defective blade would have greater-one number of defects. The upper bound of 15 was arbitrarily chosen, as densities requiring more than 15-times the number of defects would indicate some issues with the heuristics. The best results were achieved with a multiplier value of 10.

If there are regions in the input that are labelled as defective, then their suspect points are clustered with the DBSCAN algorithm [89]. These clusters now more concretely represent the actual defect and allow for the computation of their characteristics with the bonus of additional noise being removed. The DBSCAN algorithm used a neighbourhood radius of 15 and a density threshold of 3.

Renderer and Descriptor

This component takes the input image and a list of clusters found by the comparison module and computes the bounding box with padding for each cluster. The bounding box is drawn onto the input image with a unique colour and ID. The mathematical properties of the cluster are also computed with respect to the non-padded bounding box. Firstly, the absolute width and height of the defect is calculated as a function of the max and min values for the x- and y-coordinates of all points in a cluster. Secondly, the area in square pixels is calculated with respect to the polygon formed by the points. Lastly, the minimum angle is computed in relation to the interior-most point and exterior-most points with minimum and maximum y-values. Interior and exterior-most refers to points where their x-coordinate is closest and farthest to the x-coordinate of the centre of mass of the contours that make up the blade respectively. Finally, the image with the defects drawn and the list of the properties of each defect are output.

The method used for each image-processing step introduced in the previous sections and a visualisation of the results is shown in Figure 9.

| Process Step | | | | |
|---|---|---|---|---|
| 1. Input image import | 2. Grey-scaling & filtering | 3. Contour detection | 4. Feature point extraction | 5. Output image with defect bounding box |
| **Visualisation of Processing Step** | | | | |
| | | | | |
| **Method** | | | | |
| Load and scaling source image with the factor of 0.3 | RGB to grey conversion & bilateral (Shift-invariant Gaussian) filter | Canny edge detector and contour extraction using Suzuki algorithm | Contour following and angle thresholding. | Comparison with ground-truth model, point clustering with DBSCAN and drawing of bounding box |

**Figure 9.** Current image processing procedure and defect detection output.

### 3.2.3. Decision Support Tool

The decision support tool (DST) starts from the output of the detection software and applies engine maintenance heuristics to determine the serviceability of the blade. The rule-based approach uses a lookup table to retrieve the data (limits) and compare it to the findings of the detection software. The look-up table includes the limits for each different blade stage and zone. These limits are fictional numbers for reasons of commercial sensitivity. The purpose is to prove the concept rather than develop a commercial, ready-to-use software. The performance of the decision support tool was measured using the true and false decision outputs. This was done by comparing the maintenance decisions of the decision support tool with the ground truth that was determined by a senior inspector with over 30 years of experience in the field.

In a first step, we developed a reference table to record all the relevant information and measurements (Figure 10). The table was structured the following way: In the first column, the blade stages were listed and each of them was further sub-divided in the second column into the three blade zones A, B and C. These zones describe regions in which the same inspection limits apply. They are defined by their location on the blade, expressed by a set of the x/y-coordinates (column 3 to 6). For each zone, three defect size limits are listed (column 7 to 9). These contain an acceptance-, repair- and reject-threshold.

For ease of processing, the zones are measured in pixels and are determined by a set of x/y-coordinates that represent the top left and bottom right corner of each zone area. The origin of the coordinate systems is at the top left corner, with the *x*-axis pointing to the right and the *y*-axis pointing downwards. The coordinate system is shown in Figure 11.

The user interface is divided into three sections: input from the output file of the detection software, manual input required by the operator, and the decision result (Figure 12).

The output file of the detection software contains the defect location, dimensions and shape descriptors (defect characteristics). However, not all of this information is needed for the DST and only the required data is extracted. This includes the defect location and the depth of the damage.

| Stage | Zone | x-coordinate [pixel] | | y-coordinate [pixel] | | Defect size [mm] | | |
|---|---|---|---|---|---|---|---|---|
| | | start | end | start | end | accepted | repaired | rejected |
| 6 | A | 41 | 424 | 0 | 208 | ≤ 0.8 | ≤ 2 | > 2 |
| | B | 41 | 433 | 209 | 484 | ≤ 0.4 | ≤ 1.5 | > 1.5 |
| | C | 43 | 433 | 485 | 692 | ≤ 0.1 | ≤ 1.3 | > 1.3 |
| 7 | A | 43 | 384 | 0 | 155 | ≤ 0.8 | ≤ 2.0 | > 2.0 |
| | B | 47 | 383 | 156 | 362 | ≤ 0.4 | ≤ 2.0 | > 2.0 |
| | C | 54 | 381 | 363 | 517 | ≤ 0.1 | ≤ 0.7 | > 0.7 |
| 8 | A | 19 | 371 | 0 | 190 | ≤ 0.8 | ≤ 1.7 | > 1.7 |
| | B | 20 | 374 | 191 | 442 | ≤ 0.4 | ≤ 1.7 | > 1.7 |
| | C | 22 | 376 | 443 | 632 | ≤ 0.1 | ≤ 1.3 | > 1.3 |
| 9 | A | 17 | 345 | 0 | 166 | ≤ 0.8 | ≤ 1.5 | > 1.5 |
| | B | 19 | 355 | 167 | 386 | ≤ 0.4 | ≤ 1.5 | > 1.5 |
| | C | 21 | 361 | 387 | 552 | ≤ 0.1 | ≤ 0.8 | > 0.8 |
| 10 | A | 34 | 354 | 0 | 103 | ≤ 0.8 | ≤ 1.3 | > 1.3 |
| | B | 31 | 363 | 104 | 239 | ≤ 0.4 | ≤ 1.3 | > 1.3 |
| | C | 28 | 365 | 240 | 342 | ≤ 0.1 | ≤ 0.5 | > 0.5 |
| 11 | A | 23 | 361 | 0 | 97 | ≤ 0.8 | ≤ 1.3 | > 1.3 |
| | B | 21 | 370 | 98 | 226 | ≤ 0.4 | ≤ 1.3 | > 1.3 |
| | C | 19 | 373 | 227 | 323 | ≤ 0.1 | ≤ 0.8 | > 0.8 |
| 12 | A | 35 | 362 | 0 | 98 | ≤ 0.8 | ≤ 1.3 | > 1.3 |
| | B | 28 | 379 | 99 | 230 | ≤ 0.4 | ≤ 1.3 | > 1.3 |
| | C | 27 | 378 | 231 | 328 | ≤ 0.1 | ≤ 0.8 | > 0.8 |

**Figure 10.** Blade zones and defect limits reference table.

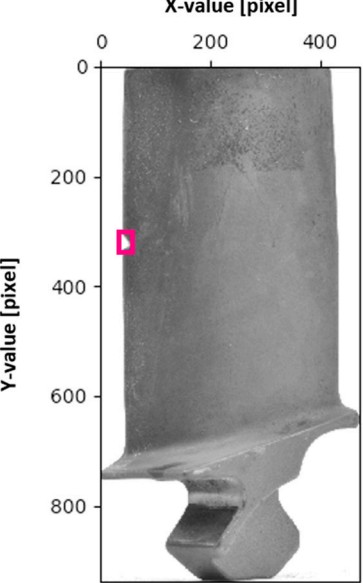

**Figure 11.** Image coordinate system with origin at top left corner.

The x- and y-coordinates of the defect location are used to determine in which blade zone the defect is located. This is important as each zone has different limits in terms of allowed damage size. The detection software delivers a set of two x/y-coordinates that define the bounding box of the detected defects. We took a risk adverse approach and therefore used the x/y-coordinates of the bottom right corner of the bounding box rather than the centre point coordinates, as former are closer to the root and thus more critical.

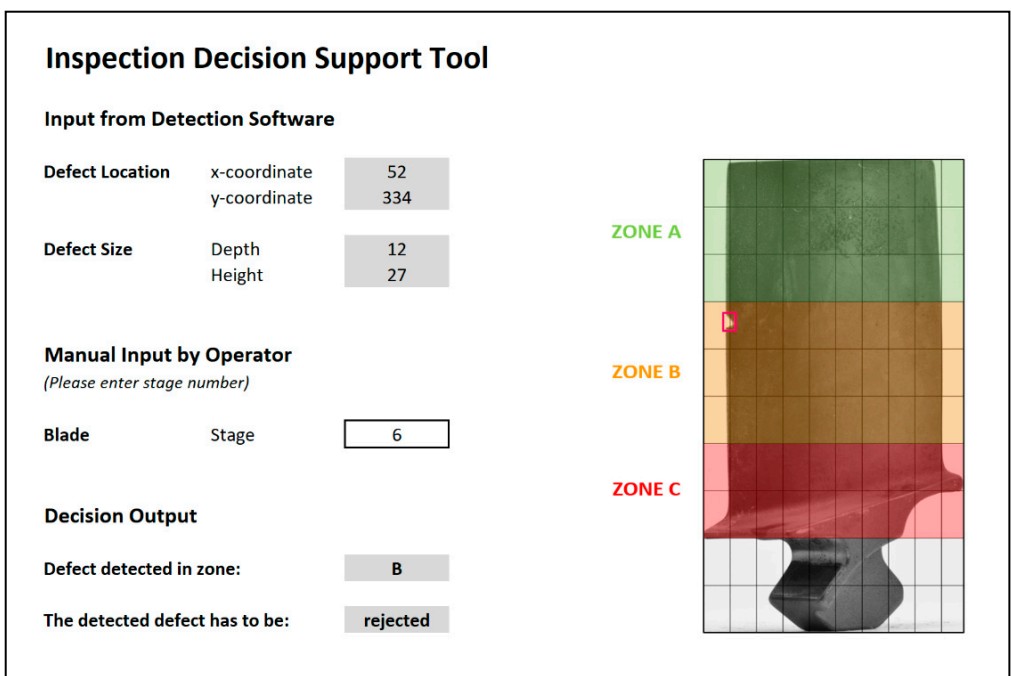

**Figure 12.** User interface of decision support tool (DST).

Since the different blade stages vary in size, the dimensions of the blade zones vary as well. Therefore, the stage number is a required input size to determine which set of limits to use. This number cannot be retrieved from the input image, as the information was not stored in the image, e.g., in the file name. Thus, it has to be manually entered by the operator.

The software then returns the identified blade zone in which the defect was detected, i.e., zone A, B or C. This interim result was needed to evaluate if the zone classification was done correctly. The classification results were compared to the senior inspector, who was given the actual part and a scale to determine the blade zone.

Next, the defect size (depth) computed by the detection software was compared to the allowed limits of the relevant zone and stage listed in the reference table. The data were interpreted as follows:

1. If the defect size is smaller or equal to the acceptable defect size, then the defect is acceptable and the blade airworthy.
2. If the defect is bigger than the acceptable defect size but smaller or equal to the reject threshold, then the defect is repairable and the blade serviceable once the airworthy condition has been retrieved.
3. If the defect size is above the reject threshold, then the defect is not repairable anymore, and the blade must be scrapped.

Depending on the comparison result, the tool then returns one of the following three decision outputs: The detected defect has to be (1) accepted, (2) repaired or (3) rejected.

## 4. Results

### 4.1. Defect Detection Software (DDS)

We performed two experiments. The first one analysed the effect of the blade perspective on the detection performance of the software. Eight models were trained with images of the according perspectives, and the best viewing angles were determined based on the true positive and false positive rate. The second experiment used the best two viewing angles and tested the model with optimal parameters to determine the accuracy of the software. These parameters were determined using an imperative approach in which the parameters that produced the best F1 metric on a small subset of the research sample was selected. A grid-search method was used to find the best parameters by running

exhaustive trials on both, the thresholding values for finding angle derivations and density threshold, as well as on the radius parameter for the DBSCAN algorithm. This produced the parameters with values as discussed in Section 3.2.2 and summarised in Figure 16.

### 4.1.1. Evaluation Metrics

The performance of the proposed software was measured based on the detection rates from the confusion matrix, which is a commonly used evaluation method for defect detection applications [90]. The ground truth was determined by an inspection expert and formed the basis of comparison between the computed and actual detections. Evaluation criteria included the probability of defect detection, namely recall rate (also called true positive rate (TPR) or sensitivity), the precision of the detection (also referred to as positive predictive value (PPV)), and the accuracy of detection based on the F1-score. The latter takes into account both the precision and recall rate. The three measures are defined as:

$$Recall = \frac{TP}{TP+FN} \times 100\%$$
$$Precision = \frac{TP}{TP+FP} \times 100\% \tag{4}$$
$$F1 - score = \frac{2 \times Precsion \times Recall}{Precision + Recall} = \frac{TP}{TP+\frac{1}{2}(FP+FN)}$$

where $TP$ represents the correct detection of a present defect in the input image; $FP$ refers to the false detection of a defect that is not present on the picture, and $FN$ describes the missed detection of a present defect.

### 4.1.2. Experiment 1

Since the introduced system applies a grid-based approach, the algorithm would cut the blade image (particularly in the edge views) into very thin slices, which may result in defects being in multiple regions. In the case where the density of suspect points is lower than the threshold, it would cause more false positives. Thus, it is important to determine the best viewing angles in order to maximize the detection rates and minimize the false positive rates when used with a larger dataset of unseen images.

First, eight different ground truth models (one for each perspective) were created by the model generator. The dataset for the model generation included 160 images of 20 non-damaged blades taken from eight different perspectives.

Subsequently, a test dataset of 104 images of eight defective and five non-defective blades from eight different perspectives each was processed. For each perspective, the performance of the model is shown in Figures 13 and 14. The viewing perspectives with the lowest incorrect detections (false positives) are one, three, seven and eight.

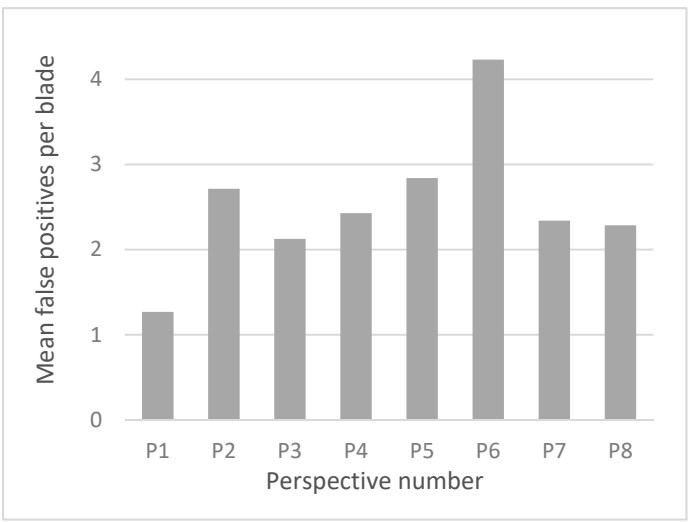

**Figure 13.** Average false positive rates for each perspective.

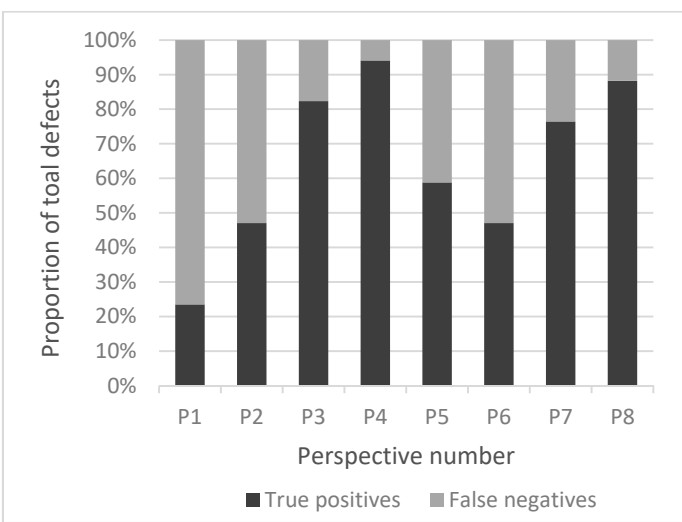

**Figure 14.** True positives and false positives proportions per perspective.

Additionally, as shown in Figure 14, the increase in false positives directly correlates with the decrease in true positives and an increase in false negatives. This experiment showed that the best viewing perspectives to use for model training are three, four, seven and eight, which are the perspectives most perpendicular to the airfoil.

### 4.1.3. Experiment 2

The second experiment tested the optimal algorithm parameters and showed the detection power of the heuristic method. The sample size for this experiment was 44 defective and 3 non-defective blades, adding up to 47 blades in total. For each of those blades, two images—one front perspective (P3) and one back perspective (P7)—were processed. As seen in Figure 12, the optimal model performs well across different sizes of blades (stages 6 to 9) from both, the front and back perspectives. This is due to the gridding feature of the detector making the models more resilient to physical size changes of the blades themselves. Overall, a TP (recall) rate of 83%, FN rate of 17% and precision of 54% were achieved across a testing dataset of 94 images. This indicates that most of the defects are being found; however, many false positives show that the increased sensitivity to defects also increased the false positives. An overall F1-score of 59% was achieved. An F1-Score of 100% means perfect accuracy and precision, whereas an F1-Score of 0% indicates that no correct detections were made. The detection performance was consistent among the different defect types. It is to be noted that detections occurring in the roots of the blades were not counted as they are excluded by the decision support tool.

The decision support tool was developed as a supplement to further improve the automated detection system by reducing the number of false positives and improving the accuracy of the results. The reduced FP rates and F1 scores are presented in Figure 15 (stacked diagrams) and further discussed in Section 4.2.2.

The algorithm parameters used for the detection software are further described in Figure 16.

### 4.2. Decision Support Tool
#### 4.2.1. Evaluation Metrics

Both location and size of defect are input variables of the decision support tool, and thus, their accuracy directly affects its performance. For instance, if the computed defect location deviates from the true location, it could consequently be allocated to a different tolerance zone, and hence, incorrect inspection limits would be applied. Likewise, if the depth were computed incorrectly, the defect might be classified as less or more critical than it actually is. This would lead to release of an unairworthy part to service, or unnecessary repair or scrapping of the blade respectively.

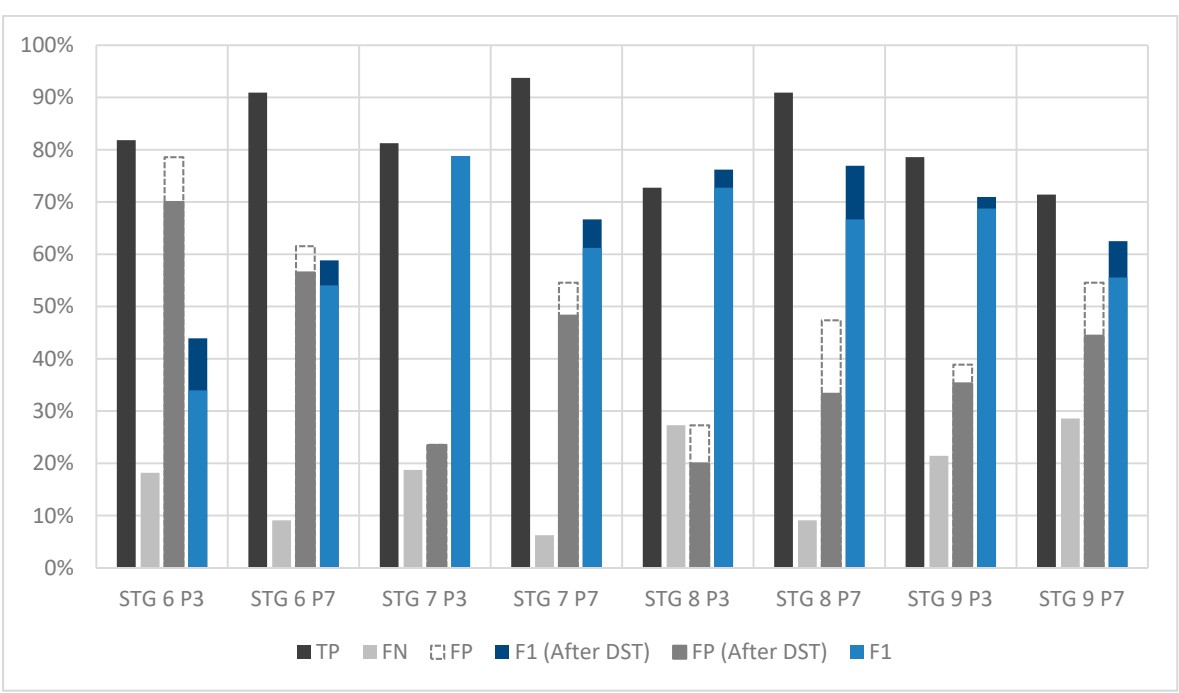

**Figure 15.** TP, FP, FN and F1-score results for stages (STG) 6 to 9 and viewing perspectives P3 and P7.

| Parameter | Description | Value |
|---|---|---|
| ANGLE_TOLERANCE | Acceptable angle threshold for two neighbouring points | 15 |
| POINT_STEP | Number of contour points to step over to produce two neighbouring points | 4 |
| AREA_FILTER_MIN | Proportion of the largest contour area in which smaller contours are a part of the blade edge | 0.05 |
| DETECT_FILTER_MIN | Multiplier for the point density required between the training model and an input model to consider a defect | 10 |
| CANNY_SIGMA | Canny edge detector $\pm$ multiplier for minimum and maximum thresholds | 0.33 |
| SCALE_FACTOR | Input image compression rate | 0.3 |
| DBSCAN_EPSILON | DBSCAN algorithm radius of nearest-neighbours inclusion | 15 |
| DBSCAN_MINPTS | Minimum number of points required to continue the cluster w.r.t the currently observed point in the DBSCAN algorithm | 3 |

**Figure 16.** Algorithm parameters.

Therefore, it was important to understand the accuracy of the defect characteristics. Evaluation metrics for defect size estimation and defect location were developed. The discrepancy in defect size was defined as the absolute error $\epsilon$ between the computed defect depth $d_c$ or height $h_c$ and the actual defect depth $d_a$ or height $h_a$, respectively. The percentage error $\bar{\delta}d$ and $\bar{\delta}h$ normalises the error based on the actual defect size, which represents the error more accurately, in particular when the defect sizes varied quite significantly. The metrics are defined as

$$\epsilon d = |d_c - d_a|$$
$$\epsilon h = |h_c - h_a|$$
$$\delta d = \left| \frac{d_c - d_a}{d_a} \right| \times 100\% \tag{5}$$
$$\delta h = \left| \frac{h_c - h_a}{h_a} \right| \times 100\%$$

Equally, the accuracy of the defect location is determined by comparison of the computed and the actual location. The resulting discrepancy is the absolute error in x-direction $\epsilon x$ and y-direction $\epsilon y$ respectively. This can also be expressed as relative location error in in x-direction $\delta x$ and y-direction $\delta y$. A radial discrepancy measure was introduced, which takes both, the displacement of the defect location in x- and y-direction into account. The radial error $\delta r$ is calculated using the relative Euclidean distance. Equation (6) describes the metrics further:

$$\epsilon x = |x_a - x_c|$$
$$\epsilon y = |y_a - y_c|$$
$$\delta x = \left| \frac{x_a - x_c}{w} \right| \times 100\% \tag{6}$$
$$\delta y = \left| \frac{y_a - y_c}{h} \right| \times 100\%$$
$$\delta r = \sqrt{\delta x^2 + \delta y^2}$$

where $x_a$ and $y_a$ are the actual x- and y-coordinates, and $x_c$ and $y_c$ are the computed coordinates of the defect location, respectively. The error in x-direction was calculated based on the blade width $w$ and based on the blade height $h$ for the discrepancy in y-direction.

### 4.2.2. Decision Output and Recommended Maintenance Action

The decision support tool relies on the output (morphology) of the detection software. The mean deviation of the computed defect location compared to the actual one was 3 pixel or 0.9% in x-direction and 8 pixel or 1.3% in y-direction. This translates into a mean error of 9 pixels or 1.6% in radial direction. The defect size had a computation error of 9 pixels or 6.3% for the depth and 35 pixels or 22.4% for the height. Therefore, the defect location was determined with 98.4% accuracy, while the defect depth estimation was 93.7% accurate. This performance of the location determination was uniform across all defect positions on both, leading and trailing blade edges. However, the percentage error tends to be bigger for shallow defects than for deeper ones. This can be explained by reviewing Equation (5). If a defect is two pixels in depth, but the software determined it to be three pixels (or vice versa), then the error rate is 50%. Whereas a large defect of 20 pixels with a one-pixel discrepancy results in a percentage error of only 5%.

The DST was only able to process positive detections made by the DDS, i.e., true positives and false positives. Figure 17 lists the computed defect characteristics with their true values and the discrepancy listed next to it. The DST then processes that information following the procedure described in Section 3.2.3 and provides a maintenance recommendation. The two right-hand columns compare the maintenance decision made by the decision support tool with the decision of the human operator. The basis for the evaluation is the ground truth that was determined by inspection experts. In doing so, they considered whether the observed condition is an acceptable or repairable defect or if the blade has to be scrapped. The engine maintenance manual provides details for this determination. The results show that the DST has recommended the correct maintenance action in most cases.

There is a small but important difference in terminology for different roles: to an inspector working in MRO, a "condition" on the blade (such as a small nick on the edge) will only be a "defect" when it exceeds a given size in a given location. In contrast, from the perspective of the detection software, any geometric anomaly on the edge is considered

a "defect". The decision-support tool encapsulates these heuristics and helps determine whether the condition is acceptable or if the defect has to be repaired or rejected.

Furthermore, the decision support tool was able to reduce the false positives by 16% by differentiating between the (true) detections that are actual edge defects and (false) detections on the root caused by the distinctive curved dovetail shape (results in Figure 15). This is done by taking the location of the computed defect and comparing it against the upper limit of zone C, which is the one closest to the root (refer to Figure 11 for image coordinate system). If the detection is located above zone C, then the finding is determined to be at the root and thus excluded from further processing.

| Defect image | Computed defect location [px] | | Actual defect location [px] | | Location error [px] | | Computed defect size [px] | | Actual defect size [px] | | Size error [px] | | Decision made by DST | Decision made by inspector |
|---|---|---|---|---|---|---|---|---|---|---|---|---|---|---|
| | $x_c$ | $y_c$ | $x_a$ | $y_a$ | $\epsilon x$ | $\epsilon y$ | $d_c$ | $h_c$ | $d_a$ | $h_a$ | $\epsilon d$ | $\epsilon h$ | | |
| | 52 | 334 | 53 | 334 | 1 | 0 | 12 | 24 | 12 | 25 | 0 | 1 | Reject | Reject |
| | 347 | 131 | 347 | 130 | 0 | 1 | 3 | 47 | 3 | 46 | 0 | 1 | Repair | Repair |
| | 342 | 36 | 362 | 50 | 20 | 14 | 27 | 30 | 47 | 50 | 20 | 20 | Reject | Reject |

**Figure 17.** Overview of computed and actual defect characteristics, and comparison of maintenance recommendations (more samples are presented in Figure A1 in Appendix A).

The detection software had particular difficulties with long smooth edge defects (such long dents), where the deformation expressed in change in angle is below the threshold and thus was not detected. A common challenge in inspecting blades for both, neural networks and image processing is the detection of large material separations (breakage) typically found at the corners (refer to Figure 5D). Software tends to struggle with those defects as the algorithm cannot detect continuation of the line [55]. The proposed system was able to detect correctly all teared-off corners. However, the computed bounding box was significantly smaller than the actual defect, which resulted in large discrepancies of the defect size and location in those few cases.

In some cases, a small (absolute) error has no impact on the decision if (a) the predicted defect location is still in the same zone and (b) the computed defect size is still below the next higher threshold. Thus, the accuracy of the decision is higher than the accuracy of the defect location and size as the DST is to some extent error-resistant. When looking at the results, it was noticeable that the defect height had a much bigger error with a mean discrepancy of 29%. However, since the defect depth is the decisive measure, an incorrect estimated defect height has no impact on the decision accuracy. Finally, if both the

computed and actual defect depth are above the upper tolerance threshold, a discrepancy has no impact on the decision, since in both cases the blade is to be rejected.

## 5. Discussion

### 5.1. Comments on the Defect Detection Software

The introduced defect detection software and decision support tool both have the potential to reduce the time spent for visual inspection of aero engine components, while improving the inspection accuracy and decision consistency across inspectors. The inspection time of the detection software was 186–219 ms per blade. In comparison, the human operator requires on average 85 s for inspecting a blade during piece-part inspection and about 3 s for borescope [91]. However, to enable the use of such software, the workflow of the MRO operations has to be adjusted. The acquisition of images is yet not part of the inspection process. If in the future blades were also photographed at the point of inspection, then hypothetically those images could be fed into a system like the proposed one and used as an independent secondary check.

The inspection of engine blades is a time-consuming and tedious process with over a thousand blades and vanes to inspect per engine. Thus, another benefit of the detection software is that it will never get tired or have performance fluctuations related to vigilance. Humans in contrast are prone to error and human factors, including but not limited to vigilance problems, fatigue, distraction and most importantly complacency. This creates the risk of missing critical defects. The proposed system provides a way to reduce this risk.

The software was able to detect *"rare"* defects. Rare can be defined in two ways: (a) defect types that are rare on compressor blades, e.g., cracks, which are more common on blades in the hot section, since heat aggravated the fatigue process. Corrosion is another uncommon type of defect on compressor blades, but since it is a less critical surface defect, a different detection approach is required. (b) Small defects can be rare since most blades with nicks and dents originate FOD engines and are quite severe. Small defects were included in the experiment. However, since the blade sample provided was relatively small, there was no rare defect type (crack) present and thus could not be tested.

The shape of the blades is an important factor, which resulted in separate models for each different perspective being trained. This is because of the non-symmetrical nature of the blades. This caused increased false positives around the roots of the blades. Due to the nature of borescope inspections, it is not always guaranteed that the front/back orientation and the stage number would be easy to discern without significant additional input from the engineer using the software. Therefore, it is required in the future to add additional filtering and logic to normalise the orientations and back/front views to appropriate models.

The main drawback of the proposed solution is the model comparison module, where only the point densities of each region are being compared. The point densities do not carry representation of the shape of the points that have been considered suspect. Therefore, the shape properties and other comparison between them cannot be done. Therein also lies an issue in which defects that are present near the edges of the regions may not be picked up as the number of suspect points would be distributed across different grid cells, thus reducing the number of points per region. This can lead to the problem of those regions having their suspect point densities falling below the required threshold and therefore contributing to a false-negative detection.

A potential solution to this issue is to perform DBSCAN clustering before generating a model and determining the average shape of a defect with the centroid of the cluster being codified in the grid cells. This would remove the issue of defects being cut off because some points are not in the correct region. In order to determine average shape, a similarity-based approach could be used. This has been done in other studies to a high level of success [92–94]. Furthermore, when comparing a defect cluster to the modelled cluster a Procrustes analysis [95] might be used to measure shape similarity between the two clusters. The dissimilarity measure can then be used to determine if an input image

contains a detected defect that has not been generalised by the model, and label it as such. This method was not fully implemented by the time the project ended, and as such, an evaluation on the performance was not possible.

In terms of the overall software solution, the use of open-source libraries and Python means that the software itself can be implemented without licensing issues in a real-world solution. Furthermore, the software solution is proof of a positive response. Should appropriate optimisation techniques be applied for the parameters, perhaps using experimental metrics as a loss function, then the performance of the software might be improved.

Future work could include evaluating the accuracy of the bounding boxes the surround a defect, as well as comparing the software performance with the human performance for the same data sets. A significant improvement would be the ability to accept video streams and perform real time processing on borescope videos. Addition of different defect profile types would allow for an increased scope on the ability to perform defect detection, as well as performing execution optimisation that would allow multiple profiles to be used in real time.

While edge defects mainly focus on the continuity of the edges of the blades, surface defects, such as corrosion, airfoil dents and scratches would require computation of surface meshes and derivations in the geometric representations of the surface as shown by Barber et al. [96]. Thus, additional image processing modules would be required to develop the current system into a workable system.

### 5.2. Comments on the Decision Support Tool

The decision support tool avoids subjectivity in the decision process and incorrect decision-making when it comes to the serviceability determination of a blade. Generally, inspectors are rather risk adverse (low risk appetite) as they know the consequences a missed defect could have. However, this can lead to costly teardowns and unnecessary scrapping of airworthy parts, which introduces a high cost to the MRO provider. Thus, the proposed method supports optimisation of the maintenance processes and operation efficiency.

The proposed method might be transferable to other levels of inspection (refer to Figure 1) such as module inspection, where the blades are still mounted on the shaft, and theoretically, an automated image acquisition tool could obtain photographs and forward them to an automated inspection system for evaluation. This would allow identifying any blades that are unserviceable (scrap) before the detailed piece-part inspection, and this may reduce the workload for the operators.

In borescope inspection, the acquisition of images or rather videos is already a well-established process. The challenge with automating this form of inspection is the inconsistent camera position and orientation, in combination with the challenging environment [8,9]. However, if the image acquisition task could be standardised, the detection system and decision support tool would have a fair chance to be applied successfully.

The DST counteracts subjective judgement of the human operator and supports moving away from a "best guess" approach towards a quantifiable and justifiable decision making process. Ultimately, this could reduce the amount of airworthy parts being scrapped, while avoiding critical defects being missed.

One limitation of the proposed decision support tool is that it treats all detected defects as dents and nicks in terms of their inspection limits. It can yet not process tears that always have to be rejected, independent of the defect size. The reason for this restriction is that the defect classification was not realised and is future work. Previous research showed that a rule-based framework could be used to classify defect types [62]. The descriptor that provides mathematical morphology introduce in Section 3.2.2 is also capable to extract additional characteristics, including information about the amount of missing material and edge deformation. This has the potential to be used for such a defect classification framework based on the characteristic appearance of the defect. Although this has not been part of this research, therein might lie the advantage that such a classification is possible even with small datasets, whereas a neural network has to be trained on hundreds

of images. Once the classification has been achieved, the decision support tool can be advanced by adding different inspection limits for each defect type. For said reasons, the detection software with its defect characteristics extraction capabilities was kept separately from the decision support tool to provide more flexibility to use the morphology for other purposes such as defect classification.

Additional contextual factors could be added to the decision support tool, based on available data on, e.g., the operational environment, engine history and previous engine shop visits and repairs. Therein lies the potential of advancing the introduced decision support tool from an appearance-based diagnosis towards a contextual-based diagnosis system.

When implementing the decision support tool in the maintenance operations environment, some considerations towards the total number of allowed defects, per blade, stage and engine have to be made. This feature was not included in the decision tool.

The defect detection software and decision support tool could be transferred to other turbo machinery and power generation applications, such as steam turbines. The blades are quite similar in shape and materials. The decision support tool could also be applied, e.g., to wind turbine blade inspection and broader inspection tasks within the manufacturing industry or to other industries such as medical examination [71].

*5.3. Performance Comparison*

The recall rate of the proposed detection software in combination with the DST was 82.7%, and the precision was 54.1%. The detection rate is comparable with the performance of neural networks in the reviewed literature [21,35,42,48], which ranged from 64.4% to 85%. Their performance was highly depended on the chosen deep learning approach and the specific requirements, i.e., what blades are being inspected, what defect types and sizes shall be detected and whether it was piece-part, module or borescope inspection. The detection rates of borescope applications were generally higher than the ones of piece-part inspection, since videos were being analysed, and a defect was present on, e.g., 50 individual frames (images). If the defect was detected in at least one frame, it was classified as TP, and thus, detection rates of 100% could be achieved [43].

Note that there is a lack of consistency when it comes to reporting the performance of inspection systems. Some researchers only reported the TP rate [35,42] or the FP and FN rates [20], while others reported the error rate [48], detection rate [21,43] or accuracy [43], and still, others made it dependent of the defect size and used a probability of detection curve [33]. For some neural networks, the performance was measured in pixel accuracy, which describes the quality of the classification rather than the detection [9,47]. This makes comparison somewhat difficult.

The detection results are also comparable to the human inspector, who has a commonly quoted error rate of 20% to 30%, or rather a detection rate of 70% to 80% [97,98]. In aircraft visual inspection in particular, the defect detection rate was stated to be 68% to 74% [42,99]. There is no inspection performance reported in the literature specifically for blade inspection. Future work could assess the detectability rates of a variety of inspectors with different experience levels by showing them the same images and make a direct comparison between the human and software performance.

When it comes to the quality of the defect size estimation, our results were comparable with other research results. An error rate of less than 13% was achieved. In comparison, Tang et al. used a Markov algorithm to predict the depth and diameter of defects of similar size (1–2.5 mm) with a percentage error of 10% [100]. There has been no work found that specifies the exact location of the detected defect. The results of Kim et al. [43] are presented in a coordinate system, so that the inspector can read of the values, but this is a manual process and has not been automated yet.

The proposed system is a first proof of concept, and the accuracy, recall rates and sensitivity can be further improved. The results indicate that image-processing techniques

can be used for small dataset and a comparable detection performance to both, neural networks and human inspectors can be achieved.

## 6. Conclusions

The objective of this project was to apply image-processing methods to underlay a decision support tool used in aircraft engine maintenance. Both are heuristic approaches, as opposed to neural networks, and both showed a promising degree of performance. As it currently stands, the software solution can use image processing and computer vision techniques to detect defects on the leading and trailing edges of compressor blades. The approach has the distinct advantage of requiring a relatively small dataset, while achieving a detection performance comparable to the human inspector and neural networks. Furthermore, there are multiple avenues for improvement: optimisation of the algorithm parameters, implementing a solution to the grid-based analysis issue and incorporating better defect profiles. Hence, we propose that image-processing approaches may yet prove to be a viable method for detecting defects.

The work also identifies which viewing perspectives are more favourable for automated perspectives, namely, those that are perpendicular to the blade surface. While this is somewhat intuitive, it is useful to have quantified the effect.

A decision support tool was proposed that provides the inspector with a recommended maintenance action for the inspected blade. The rule-based approach was proven reliable, and any inaccuracies in the decision were caused by discrepancies in the defect size and location computed by the detections software. Future work could include: incorporating different defect types and their corresponding tolerances, enhancing the decision algorithm by taking into account the findings of several blades and their dependencies, and integrating the decision support tool into the defect detection software. The proposed automated systems have the potential to improve the speed, repeatability and accuracy, while reducing the risk of human errors in the inspection and decision process.

**Author Contributions:** Conceptualization, J.A. and D.P.; methodology J.A., D.P., S.S. and R.M.; software, S.S., R.M. and A.M.; validation, J.A.; formal analysis, S.S., R.M. and A.M.; investigation, J.A. and S.S.; resources, J.A.; data curation, S.S. and R.M.; writing—original draft preparation, J.A. and D.P.; writing—review and editing, J.A., S.S., D.P., R.M. and A.M.; visualization, J.A. and S.S.; supervision, D.P., R.M. and A.M.; project administration, D.P., R.M. and A.M.; funding acquisition, D.P. and A.M. All authors have read and agreed to the published version of the manuscript.

**Funding:** This research project was funded by the Christchurch Engine Centre (CHCEC), a maintenance, repair, and overhaul (MRO) facility based in Christchurch, and a joint venture between the Pratt and Whitney (PW) division of Raytheon Technologies Corporation (RTC) and Air New Zealand (ANZ). The APC was partly funded by the UC Library Open Access Fund.

**Institutional Review Board Statement:** Not applicable.

**Informed Consent Statement:** Not applicable.

**Data Availability Statement:** The data are not publicly available due to commercial sensitivity.

**Acknowledgments:** We sincerely thank staff at the Christchurch Engine Centre for their support and providing insights into visual inspection and determining the ground truth for performance assessment of the proposed inspection tool. In particular, we want to thank Ross Riordan, Marcus Wade and David Maclennan.

**Conflicts of Interest:** J.A. was funded by a PhD scholarship funded by the Christchurch Engine Centre (CHCEC). The authors declare no other conflicts of interest.

## Appendix A

Figure A1 shows more results of the defect detection software compared to the actual defect characteristics and the recommended maintenance action of the decision support tool compared to the decision made by the human operator.

| Detection number | Computed defect location [px] | | Actual defect location [px] | | Location error [px] | | Computed defect size [px] | | Actual defect size [px] | | Size error [px] | | Decision made by DST | Decision made by inspector |
|---|---|---|---|---|---|---|---|---|---|---|---|---|---|---|
| | $x_c$ | $y_c$ | $x_a$ | $y_a$ | $\epsilon x$ | $\epsilon y$ | $d_c$ | $h_c$ | $d_a$ | $h_a$ | $\epsilon d$ | $\epsilon h$ | | |
| 1 | 25 | 161 | 38 | 156 | 13 | 5 | 3 | 33 | 3 | 27 | 0 | 6 | Repair | Repair |
| 2 | 41 | 110 | 40 | 110 | 1 | 0 | 17 | 48 | 16 | 45 | 1 | 3 | Reject | Reject |
| 3 | 346 | 112 | 346 | 100 | 0 | 12 | 20 | 45 | 20 | 32 | 0 | 13 | Reject | Reject |
| 4 | 37 | 42 | 49 | 32 | 12 | 10 | 13 | 33 | 25 | 28 | 12 | 5 | Reject | Reject |
| 5 | 35 | 152 | 36 | 117 | 1 | 35 | 12 | 64 | 11 | 29 | 1 | 35 | Reject | Reject |
| 6 | 18 | 95 | 18 | 98 | 0 | 3 | 2 | 17 | 3 | 18 | 1 | 1 | Accept | Repair |
| 7 | 343 | 31 | 343 | 30 | 0 | 1 | 23 | 30 | 23 | 30 | 0 | 0 | Reject | Reject |
| 8 | 347 | 109 | 347 | 107 | 0 | 2 | 12 | 38 | 12 | 27 | 0 | 11 | Reject | Reject |
| 9 | 31 | 169 | 6 | 28 | 25 | 141 | 6 | 28 | 7 | 32 | 1 | 4 | Repair | Repair |
| 10 | 22 | 216 | 22 | 213 | 0 | 3 | 1 | 16 | 1 | 17 | 0 | 1 | Accept | Accept |
| 11 | 350 | 167 | 350 | 167 | 0 | 0 | 10 | 34 | 10 | 33 | 0 | 1 | Reject | Reject |
| 12 | 347 | 131 | 347 | 130 | 0 | 1 | 3 | 47 | 3 | 46 | 0 | 1 | Repair | Repair |
| 13 | 350 | 164 | 349 | 162 | 1 | 2 | 15 | 84 | 12 | 88 | 3 | 4 | Reject | Reject |
| 14 | 34 | 132 | 35 | 150 | 1 | 18 | 10 | 50 | 11 | 66 | 1 | 16 | Reject | Reject |
| 15 | 347 | 95 | 347 | 94 | 0 | 1 | 4 | 19 | 6 | 19 | 2 | 0 | Repair | Repair |

**Figure A1.** Result table with computed and actual defect characteristics and generated maintenance actions compared to actual action.

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
