# Peer review of "Automated Defect Detection and Decision-Support in Gas Turbine Blade Inspection"

_aerospace, doi:10.3390/aerospace8020030_

Round 1

Reviewer 1 Report

This paper reports on an automated defect detection system and decision support tool for aircraft gas turbine blade inspection. While the common approach in the field consists mainly of AI/Deep Learning, this work relies on traditional Computer Vision methods in order to bypass the burden related to the gathering of large training datasets of a specific target.

  1. One major point of focus of this paper is about small training datasets. The notion of small vs large is however subjective and varies between fields and applications. Additional discussion on this matter would be welcome. Is the dataset considered in the experiment considered small? If authors could provide some more insights for the range of applicability of the proposed method, i.e., when it is suitable compared to a Deep Learning approach and vis-versa, would benefit the paper.
  2. From my reading of this paper, it is my understanding that the proposed method is to be used on disassembled fan blades. However, the mention of borescope inspections and inspection of assembled turbines starting in line 591 do cause confusion. Providing additional details regarding the inspection workflow with the proposed method at the beginning of the paper would increase the clarity of the paper.
  3. What are the advantages of the proposed method compared to human inspection? Since fan blades are considered to be disassembled, the main advantage would be time savings during the actual visual inspection? How long does it take for a human inspector to visually inspect a fan blade?
  4. In line 101, the authors point out a previous work’s issue of only being able to detect a single type of defect. Additional details to reinforce this point would be welcome. Additionally, if coverage of several types of defects is important, a discussion on this point in the experiments would be needed Which defects in what proportions were included in the experiments? Were there notable performance gaps of the proposed method depending on the defect type?
  5. In line 185, the authors state the importance of detecting rare defects (in line 66 it is stated that dents, nicks, and tears are common defects). What are such defects? Were they present in the experiments?
  6. The threshold parameter mentioned in line 307 seems paramount to the performance of the proposed method. A discussion on the selection of its value is needed. The same comment applies to the threshold mentioned in “4.1.5 Comparison Model”.
  7. Experiment 1 is dedicated to determining which perspective is the most suited for defect detection. It seems intuitive that the front and back perspectives would be the most suited ones. Is there a specificity of turbine blades that made this experiment/consideration needed?
  8. In line 396, the authors mention "optimal algorithm parameters". How where those parameters are selected is an important step since they directly impact performance. A clear and detailed description, including the logic behind their selection, should be added.
  9. The experiments are lacking comparison with other methods and do not allow to assess the performance of the proposed method fairly. If the implementation of a previous work is not possible, it would be still desirable to show the performance of a basic method, such as SIFT with a simple classifier such as SVM, next to the performance of the proposed method to provide readers with an idea for a baseline performance on the considered dataset.  
  10. In section 4.3.4, the proposed method boasts high performance for defect location and depth estimation. Was this performance uniform across all defect positions and depths?

Minor comments:

  1. There is a typo in line 360
  2. Large portions of the Result section are dedicated to the description of the method, moving those to the Method section would benefit the readability of the paper.

Author Response

We thank the reviewer for the efforts and the valuable feedback.

All issues raised by the reviewers were addressed. Please refer to the attached file for a detailed list of all changes made according to your feedback.

Any changes to the manuscript are shown in red ink.

Reviewer 2 Report

Overall, this paper investigates Gas Turbine Blade Inspection by image processing. In general, there are many places that could be greatly improved, and these improvements could make this paper even better, as suggested below. The sample of this study is quite special and has research value, but the presentation of this paper must be improved to a more professional presentation.

Suggestions:

  1. It is suggested to add a red box on the top of Fig. 1 to recognize the location.
  2. Figure 5 shows that there is a problem with the light source setting of the collected samples, can it be improved? If not, can you prove that this process can be overcome?
  3. The paper mentions the use of JPEG images which are compressed, what is the compression ratio used? Please provide the compression ratio.
  4. Figure 6 illustrates the steps of the process proposed in this study, but the detailed setting parameters of this process are not found in the pictures and the text. For example, the steps mentioned in the text do not all correspond to the description of the image, and Image preprocessor is used to represent a step in the figure, but the wording is quite extensive, and the characteristics of this study are in this block, so it is recommended that the relevant setting parameters must be added, and the steps are described in more detail, including redrawing the flow diagram.
  5. The equation after "Canny edge detector, with the thresholds" in the text should be corrected and the actual used parameters should be given.
  6. The content of the subsection Analysis of Regions of Interest is difficult to understand what the author is trying to say if viewed from an imaging professional background. Please modify.
  7. In the model generator subsection, it is mentioned that CSV files are generated, which is a strange thing. As this paper says 'This matrix is then written to a CSV file', this matrix should be the weight of a model, and if so, then the name of the model should be written. And this subsection should be modified.
  8. In the Comparison Module subsection, it is difficult to understand what Model xy is? Please add a description or modify it.
  9. The presentation of Figure 7 is quite good, but also corresponds to my previous question, the lack of details is very confusing, for example, what is used for filtered? Which method is used for Contour detection? How does Feature point extraction do? (Is it according to the angle to determine as mentioned before?)
  10. In 4.2.2. Experiment 1 subsection, it is mentioned that P1, P3, P7, P8 is the lowest incorrect detections. This part of the light source shooting, are very dim, is it related to this? If so, the light source problem should be solved. If not, why are they all just happen to be the darker part of the light source?
  11. Figure 13 should be " image coordinates system"

Author Response

(The authors gave the same response as above.)

Reviewer 3 Report

The paper presents a machine vision scheme for defect detection in Gas turbine blades. The proposed method mainly focuses on edge defects (dent, nick and tear), but not surface defects in the blade image.

For each blade under inspection, images from eight different perspectives were taken. Canny edge detection and contour following are the main processes of the proposed method for defect detection. The manuscript does not present novelty in terms of methodology or application for defect detection in blade images. It is not suitable for the publication in the journal.

  1. The authors criticize that the deep learning approach is not suitable for their problem since it requires a large dataset for the model training. But, it is not true for the problem studied in the manuscript. In the manufacturing site, the environment to capture the object image can be well controlled. A few physical objects are only required, and the training samples can be obtained by augmentation.
  2. The proposed method relies on the extraction of the blade contour in the image by using Canny edge detection and contour following. The Canny edge detector could result in broken edges along the object’s outer boundary. How the contour following is carried out? 
  3. Why did the authors use edge detection to detect blade defects? Won’t it be easier by sensing the blade object with backlighting and then thresholding the high-contrast image to obtain the silhouette of the blade?
  4. It is not clear whether the proposed method requires a template for the comparison. In eq. (1), a, b, c and d are not defined.

Author Response

(The authors gave the same response as above.)

Round 2

Reviewer 1 Report

The authors have explained some of the questions from the last review. I believe the paper can be accepted now.

Reviewer 2 Report

It's okay.